# Double-Ended Synthesis Planning with Goal-Constrained Bidirectional Search

**Kevin Yu**
MIT
kyu3@mit.edu

**Jihye Roh**
MIT
jroh99@mit.edu

**Ziang Li**
Georgia Tech
ziang@gatech.edu

**Wenhao Gao**
MIT
whgao@mit.edu

**Runzhong Wang**
MIT
runzhong@mit.edu

**Connor W. Coley**
MIT
ccoley@mit.edu

## Abstract

Computer-aided synthesis planning (CASP) algorithms have demonstrated expert-level abilities in planning retrosynthetic routes to molecules of low to moderate complexity. However, current search methods assume the sufficiency of reaching arbitrary building blocks, failing to address the common real-world constraint where using specific molecules is desired. To this end, we present a formulation of synthesis planning with starting material constraints. Under this formulation, we propose Double-Ended Synthesis Planning (DESP), a novel CASP algorithm under a *bidirectional graph search* scheme that interleaves expansions from the target and from the goal starting materials to ensure constraint satisfiability. The search algorithm is guided by a goal-conditioned cost network learned offline from a partially observed hypergraph of valid chemical reactions. We demonstrate the utility of DESP in improving solve rates and reducing the number of search expansions by biasing synthesis planning towards expert goals on multiple new benchmarks. DESP can make use of existing one-step retrosynthesis models, and we anticipate its performance to scale as these one-step model capabilities improve.

## 1 Introduction

Synthesis planning—proposing a series of chemical reactions starting from purchasable building blocks to synthesize one or more target molecules—is a fundamental task in chemistry. For decades, chemists have approached the challenge of synthesis planning with *retrosynthetic analysis* [1, 2], the strategy by which a target molecule is recursively broken down into simple precursors with reversed reactions. In recent years, advances in machine learning have enabled a multitude of computer-aided synthesis planning (CASP) algorithms [3–6] that navigate a combinatorially large space of reactions to propose chemically sensible routes to a variety of drug-like molecules within seconds to minutes. However, fully data-driven algorithms still underperform when generalizing to realistic use cases such as planning for more complex targets or in constrained solution spaces. In practice, expert chemists may plan syntheses with specific starting materials in mind, called "structure-goals" [1], that constrain the solution space. For instance, efficient syntheses of highly complex drugs are often most practical when synthesized from naturally-occurring starting materials that share complex features with the target, a practice known as "semi-synthesis" [7, 8]. There is also immense interest in identifying pathways from waste or sustainable feedstocks to useful chemicals [9–11], but existing methods have thus far relied on heuristics and brute-force enumeration of reactions.

Though algorithms for planning synthetic routes from *expert-specified starting materials* have been proposed [12, 13], the vast majority of CASP algorithms today cannot address starting material-

38th Conference on Neural Information Processing Systems (NeurIPS 2024).

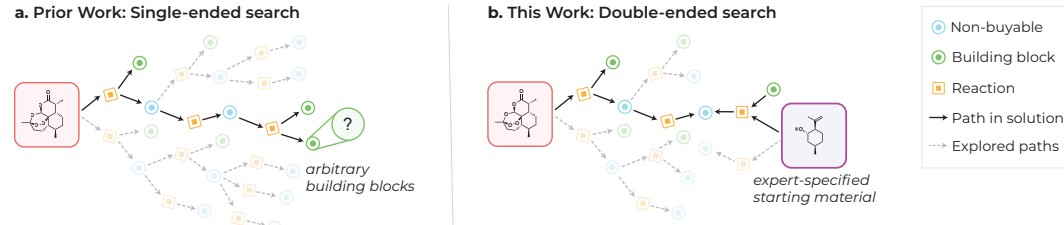

Figure 1: **(a)** Existing search methods are single-ended, and aim to identify a synthetic route where all leaf nodes meet certain termination criteria, e.g., buyability. **(b)** DESP is a bidirectional search algorithm that enables a double-ended starting material-constrained search, better reflecting certain real-world use cases in complex molecule synthesis planning. Empirically, double-ended search finds starting material-constrained solutions with fewer node expansions.

constrained use cases, as they assume that solution states may comprise any combination of building blocks. It is non-trivial to extend from "general" retrosynthesis planning to the constrained setting; by requiring a solution to contain a specific goal molecule, starting material-constrained synthesis planning presents the challenge of simultaneously guiding a search towards this goal molecule and any other necessary buyable molecules.

In this paper, we address these challenges by proposing a strategy for starting material-constrained synthesis planning with a *bidirectional search algorithm* and a goal-conditioned cost network learned offline from expert trajectories implicit to a validated reaction corpus. Given a target molecule and one or more specified starting materials, our Double-Ended Synthesis Planning (DESP) algorithm takes advantage of the natural reversibility of retrosynthesis by instantiating two AND-OR search graphs and alternately performing retrosynthetic expansions and forward synthetic expansions. We present two variations of DESP based on front-to-end (F2E) and front-to-front (F2F) bidirectional search. In F2E search, each direction of the search is conditioned on the root node of the opposing search graph, while in F2F, each search is conditioned on the "closest" nodes of the opposing search graph. In both cases, finding solutions is accelerated when the "bottom-up" search graph converges with the "top-down" retrosynthesis search graph. Each step of selection and expansion of bottom-up nodes is conditioned on a specific molecule in the retrosynthetic graph, and we devise a means of utilizing both our goal-conditioned cost network and an existing cost network for general retrosynthesis in the top-down search policy. The goal-conditioned cost network, which we term the "synthetic distance" network, is trained offline based on the observation that multi-step synthetic routes can be interpreted as expert plans where any of the non-root molecules represents a starting material goal for the final target molecule, thus bypassing the need for self-play using reinforcement learning (RL). In order to train the network on "negative experiences", we also sample pairs of molecules between which no path exists through known reactions. **Our contributions can be summarized as follows:**

- We provide a formulation of starting material-constrained synthesis planning and release the first benchmarks for evaluating algorithms on this task, including new real-world benchmarks collected from the Pistachio database [14] addressing redundancies in the widely-used USPTO-190 test set.

- We present a starting material-constrained neural bidirectional search algorithm to tackle double-ended synthesis planning. Specifically, we present a cost network that estimates the "synthetic distance" between molecules (instead of the distance between a molecule and arbitrary purchasable building blocks) and an A*-like bidirectional search algorithm that strictly reflects the constraints.

- We present strong empirical results that illustrate the efficiency of double-ended synthesis planning. Compared to existing algorithms, DESP expands fewer nodes and solves more targets under goal constraints, demonstrating its value in biasing CASP algorithms towards expert goals.

## 2 Background

### 2.1 Related work

**Computer-aided retrosynthetic analysis** Retrosynthetic analysis has traditionally been formulated as a tree search problem, where each step involves searching for chemically feasible transformations and corresponding reagents to derive the product molecule. In defining the feasible transformation,

template-based methods select graph transformation rules to apply based on expert rules [15] or use data-driven methods [16–18], such as a neural network policy trained on a reaction corpus [19]. Template-free methods frame one-step retrosynthesis prediction as a translation task of SMILES strings [20, 21] or a graph-edit prediction [22]. In searching for multi-step synthetic pathways, the focus of late has been on selecting non-terminal nodes for expansion. Initial efforts relied on expert-defined rules and heuristics [2, 15], whereas more recent efforts combine neural networks and Monte Carlo Tree Search (MCTS) [3], as well as AND-OR graph searches that address the hypergraph complexity of reaction routes [23, 6, 4, 24]. Notably, Chen et al. [6] proposed Retro*, a neural-guided A*-like search algorithm that we build on as part of our approach. Much additional work has been done to improve multi-step CASP algorithms [25–32], primarily via improvements of single-step policies in a multi-step context or value functions for improved search guidance. Unlike DESP, these methods do not address the problem formulation where the pathway search is constrained by one or more starting materials, as shown Fig. 4. One exception is GRASP [13], which utilizes RL with hindsight experience replay [33] for goal-conditioned value estimation. Additionally, starting material-oriented planning capablities were implemented in the LHASA program [12] but rely entirely on expert-defined rules. In this work, we instead train a cost network **offline** using historical reaction data and use **bidirectional search** to augment the retrosynthesis planner.

**Synthesizable molecular design**    Recent advances in computer-aided molecular design have introduced novel approaches to synthesis planning. To ensure high synthetic accessibility of designed molecules, researchers have proposed assembling compounds *in silico* by applying valid chemical transformations to purchasable building blocks, effectively searching for molecules within a reaction network [34–39]. The advent of deep generative modeling has further enabled the generation of synthetic pathways with neural models [40–44]. These methods commonly employ a bottom-up strategy, constructing synthetic pathways from building blocks to the final product. Gao et al. [42] proposed that this framework could enable "bottom-up synthesis planning," in which the goal of generation is to match a specified target molecule, and demonstrated the successful application of this approach despite a low empirical reconstruction rate. In this work, we build upon Gao et al. [42]'s method of conditional synthetic route generation by increasing the number of reaction templates, training on a larger reaction corpus, and integrating the models into a bidirectional search algorithm.

**Bidirectional search**    Bidirectional search is a general strategy that can accelerate search in problems that involve *start* and *goal* states by interleaving a traditional search from the start state and a reverse search from the goal state [45], usually guided with either neural networks or expert heuristics. It has demonstrated utility in problems such as robotic path planning [46, 47], program synthesis [48], traffic management [49], and puzzle solving [50]. However, the application of bidirectional search in synthesis planning has not been explored. When integrating an informed method of evaluating nodes, bidirectional search can be divided into *front-to-end* (F2E) and *front-to-front* (F2F) strategies [51, 52]. In F2E search, evaluations are made by estimating the minimal cost of a path between a frontier node and the opposite goal, while in F2F search, evaluations are made by estimating the minimal cost of a path between opposing frontier nodes. In this work, we implement both F2E and F2F variants of DESP to observe the empirical differences between the strategies in the synthesis planning setting.

## 2.2   Formulation of general and starting material-constrained synthesis planning problems

**General synthesis planning**    In this work, we only consider template-based retrosynthesis methods, though any single-step model is compatible with our algorithm. Let $\mathcal{M}$ be the set of all valid molecules, $\mathcal{R}$ be the set of all valid reactions, and $\mathcal{T}$ be the set of all valid reaction templates. A *reaction* $R_i \in \mathcal{R}$ is a tuple $(r_i, p_i, t_i)$, comprising a set of reactants $r_i \subset \mathcal{M}$, a single product $p_i \in \mathcal{M}$, and a retro template $t_i \in \mathcal{T}$. A *retro template* $t$ is a function $t : \mathcal{M} \rightarrow 2^{\mathcal{M}}$ that maps a product to precursors such that $\forall i : r_i \in t_i(p_i)$. Likewise, a *forward template* $t' \in \mathcal{T}'$ is a function $t' : 2^{\mathcal{M}} \rightarrow \mathcal{M}$ where $\forall i : p_i \in t'(r_i)$.

Given target molecule $p^* \in \mathcal{M}$ and set of building blocks (BBs) $\mathcal{B} \subset \mathcal{M}$, synthesis planning finds a *valid synthetic route*—a set of reactions $S = \{R_1, \dots R_n\}$ that satisfies the following constraints.

**Constraint 1** (Synthesize all non-BBs). $\forall i : m \in r_i, m \notin \mathcal{B} \implies \exists j$ s.t. $m = p_j$;

**Constraint 2** (Target is final molecule synthesized). $\exists i$ s.t. $p_i = p^*, \forall i : p^* \notin r_i$;

Finally, we require that the graph formed by $S$ is a directed acyclic graph (DAG), where for each $i$, the product $p_i$ is mapped to a node, which has a directed edge to a node mapping to reaction $R_i$, which in turn has directed edges to nodes mapping to the reactants $r_i$.

**Starting material-constrained synthesis planning**  Given a specific starting material (*s.m.*) $r^* \in \mathcal{M}$, in addition to Constraint 2, a valid synthetic route satisfies the following constraints.

**Constraint 3** (*s.m.* used). $\exists i$ s.t. $r^* \in r_i$,  $\not\exists j$ s.t. $r^* \in p_j$;

**Constraint 4** (*s.m.* not necessarily BB). $\forall i : m \in r_i, m \notin \mathcal{B} \cup \{r^*\} \implies \exists j$ s.t. $m = p_j$;

Fig. 1b illustrates an example of a valid starting material-constrained route found through bidirectional search. DESP can also be used for the more general form of the problem in which a set of potential starting materials $\{r_1^*, \dots r_n^*\}$ is given on input, and at least one leaf node must map to $r_i^*$ for some $1 \leq i \leq n$. For simplicity, we only consider the single $r^*$ case unless otherwise specified.

## 3   Methods

DESP is built on the Retro* algorithm [6] and recent advances that enable conditional generation of synthetic routes from the bottom up [41, 42].

### 3.1   Definition of synthetic distance, a goal-conditioned cost function

Like Retro* [6], DESP is an A*-like search and thus requires a method of evaluating the expected cost of various frontier nodes. We follow Retro* and use the notation of $V_t(m|T)$, $V_m$, and $rn$ functions (Section A.2 details Retro* and these functions). We also define a function $c : \mathcal{R} \to \mathbb{R}$ which maps a reaction to a scalar cost. For a valid synthetic route $S = \{R_1, \dots, R_n\}$, the *total cost* of $S$ is $\sum_{i=1}^n c(R_i)$. $V_m$ represents the minimum total cost across every valid synthetic route to molecule $m$, and is learned in Retro* and DESP to bias the search towards $\mathcal{B}$.

However, to maintain consistency in guiding A* search in the starting material-constrained setting, we require not only an estimate of the cost of synthesizing molecule $m$ from arbitrary building blocks, but also an estimate of the **cost of synthesizing molecule $m_2$ from $m_1$ specifically** (in addition to other arbitrary building blocks). As such, we define a new function $D : \mathcal{M} \times \mathcal{M} \to \mathbb{R}$, which we term *synthetic distance*, as it effectively represents the minimum cost distance between two molecules in $\mathcal{G}$, the graph constructed from the set of all possible reaction tuples $\mathcal{R}$. More precisely, the synthetic distance from $m_1$ to $m_2$ is the difference between the minimum cost of synthesizing $m_2$ across all valid synthetic routes containing $m_1$ and the minimum cost of synthesizing $m_1$ across all valid synthetic routes in general. Learning $D$ then allows for the guidance of both top-down search towards the starting material and bottom-up search towards the target with rapid node comparisons.

### 3.2   DESP **algorithm overview**

In practice, synthesis planning problems are generally approached by simulating a search through the complete reaction graph $\mathcal{G}$. We follow Xie et al. [30] in considering an AND-OR graph structure for search graphs, in which molecules are represented by OR nodes (only one child must be solved) and reactions are represented by AND nodes (all children must be solved). In implementing most synthesis planning algorithms [3, 6], one initializes the search graph $G = \{p^*\}$. With DESP, we instead initialize two search graphs $G_R = \{p^*\}, G_F = \{r^*\}$ and introduce two expansion policies, one for "top-down" retrosynthesis expansions on $G_R$ and another for "bottom-up" forward expansions on $G_F$. This allows us to perform a **bidirectional graph search** between the target and goal molecules by interleaving retro and forward expansions, with the goal of the two search graphs converging to more efficiently find a valid synthetic route. In this work, we implement F2E and F2F variants of DESP. Notably, our implementation of F2F performs node comparisons to **all nodes** in the opposing search graph rather than just frontiers. For $m \in G_R$, we define a *goal function* $\gamma : \mathcal{M} \to \mathcal{M}$ such that $\gamma(m) = r^*$ in F2E and $\gamma(m) = \arg\min_{g' \in G_F} D(g', m)$ in F2F. Likewise, for $m \in G_F$, let $\gamma(m) = p^*$ in F2E and $\gamma(m) = \arg\min_{g' \in G_R} D(m, g')$ in F2F.

The following quantities or functions are relevant in the algorithm: $rn$, $V_t(m|G)$, and $V_m$ from Retro*, and somewhat analogously $dn$, $D_t(m|G_R)$, and $D_m$. We briefly define the new quantities: **(1)** $D_m$ represents $D(\gamma(m), m)$. **(2)** $dn(m|G_R)$ represents the "distance numbers" of a top node $m$.

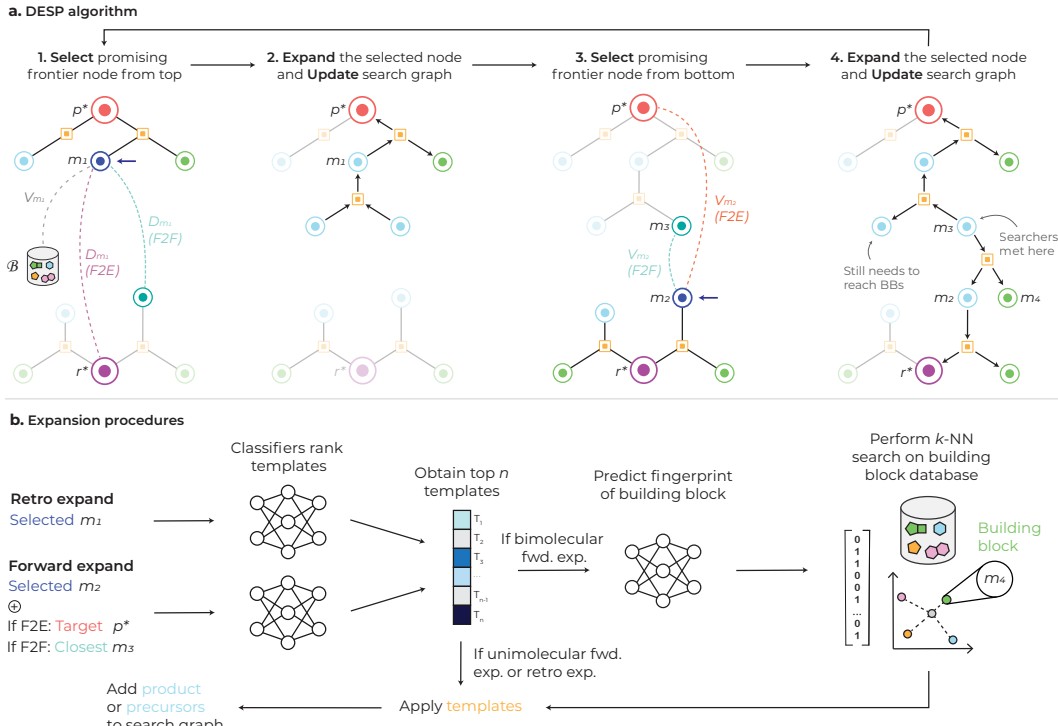

Figure 2: **(a)** DESP algorithm. Evaluation of top nodes is based on both $V_m$ and $D_m$. For F2E search, synthetic distance is calculated between a molecule and the opposing goal, while for F2F, it is calculated based on the closest opposing molecule. **(b)** Overview of the one-step expansion procedures.

This is a multiset of values $D_m - V_m$ for related frontier nodes collected for dynamic programming from the bottom-up during the update phase. **(3)** $D_t(m|G_R)$ is a multiset of all values of $D_m - V_m$ across frontier nodes in the minimum cost synthetic route to the target $p^*$ through molecule $m$. At a high level, we introduce these quantities and new policies to account for the fact that only one subgoal in a valid synthetic route needs to reach $r^*$. The top-down searcher of DESP is thus an extension of Retro* that simultaneously utilizes general retrosynthesis and synthetic distance cost networks.

Like most CASP algorithms, DESP cycles between steps of **selection**, **expansion**, and **update** until the termination criteria are satisfied. However, DESP also alternates between performing these steps for the top-down and bottom-up search graphs (Fig. 2), with each search having its own policies.

**Selection**     For top-down selection, we select an frontier molecule node that minimizes the expected total cost of synthesizing the target $p^*$ from the goal molecule $r^*$ through that node. Let $\mathcal{F}_R$ represent the set of frontier top molecules and $\mathcal{F}_F$ represent the set of frontier bottom molecules. Then,

$$m_{select,R} \leftarrow \arg\min_{m \in \mathcal{F}_R} [V_t(m|G_R) + \min(D_t(m|G_R))] \tag{1}$$

The bottom-up selection policy is identical to that of Retro*.

$$m_{select,F} \leftarrow \arg\min_{m \in \mathcal{F}_F} V_t(m|G_F) \tag{2}$$

**Expansion**     For top-down expansion, we follow other AND-OR-based algorithms in calling a single-step retrosynthesis model, applying the top $n$ predicted templates to the selected node and adding the resulting reactions and their precursors as nodes to the graph. For each added molecule node $m_i$, we initialize: **(1)** $rn(m_i|G_R) \leftarrow V_{m_i}$, equal to the Retro* value function, and **(2)** $dn(m_i|G_R) \leftarrow \{D_{m_i} - V_{m_i}\} = \{D(\gamma(m_i), m_i) - V_m\}$.

For bottom node $m$, we perform the forward expansion procedure detailed in Section 3.3, conditioned on $\gamma(m)$. For each added product $p_i$, we then initialize $rn(p_i|G_F) \leftarrow V_{p_i} = D(p_i, \gamma(p_i))$

**Update** For $G_R$, we propagate updates to relevant values up the graph and then back down to related nodes, similar to other AND-OR algorithms. As the update rules for the Retro* quantities are the same, we only provide the update rules for the new quantities, and details of the Retro* updates is in Section A.2. $G_F$ is also updated according to the Retro* algorithm (as branching from multiple product OR nodes is not allowed in forward expansions), so the following new updates only apply to $G_R$. We first uppropagate, performing updates up the graph for reaction (AND) nodes $R$ and molecule (OR) nodes $m$, where the $ch$ and $pr$ functions return the children and parent nodes for an input node:

$$dn(R|G_R) \leftarrow \sum_{m \in ch(R)} dn(m|G_R) \tag{3}$$

$$dn(m|G_R) \leftarrow \begin{cases} [D_m - V_m] & \text{if } x \in \mathcal{F}_R \\ dn\left(\arg\min_{R \in ch(m)} rn(R)\Big|G_R\right) & \text{otherwise} \end{cases} \tag{4}$$

We then downpropagate the following updates:

$$D_t(R|G_R) \leftarrow dn(pr(R)|G_R) - dn\left(\arg\min_{R' \in ch(pr(R))} rn(R'|G_R)\Big|G_R\right) + dn(R|G_R) \tag{5}$$

$$D_t(m|G_R) \leftarrow D_t\left(\arg\min_{R \in pr(m)} rn(R|G_R)\Big|G_R\right) \tag{6}$$

Justification for the rules and additional details, including DESP pseudocode, is in Section A.5.

### 3.3 Forward expansion policy with conditional generation of one-step reactions

To perform *forward one-step synthesis expansions*, we adapt the approach of Gao et al. [42]. Let $z_m^n : \mathcal{M} \to \mathbb{R}^n$ and $z_t^n : \mathcal{T} \to \mathbb{R}^n$ be functions mapping a molecule and template (respectively) to $n$-dimensional embeddings. We define two target functions:

$$f_t : \mathcal{M} \times \mathcal{M} \to \mathcal{T}' \approx \sigma(\text{MLP}_t(z_m^n(m_1) \oplus z_m^n(m_2))) \tag{7}$$

$$f_b : \mathcal{M} \times \mathcal{M} \times \mathcal{T}' \to \mathcal{B} \approx \text{k-NN}_\mathcal{B}(\text{MLP}_b(z_m^n(m_1) \oplus z_m^n(m_2) \oplus z_t^n(t'))) \tag{8}$$

Together, the learned approximations of $f_t$ and $f_b$ define our forward expansion policy (Algorithm 1), which generates forward reactions for the expanded node $m_1$ conditioned on $m_2$.

---
**Algorithm 1:** FORWARD_EXPAND($m_1, m_2, G_F, N, K$)

$m_1$: molecule selected for expansion, $m_2$: molecule to condition expansion on, $G_F$: bottom search graph, $N$: num. templates to propose, $K$: num. building blocks to search

---
$t' \leftarrow \text{TOP\_N}(\sigma(\text{MLP}_t(z_m(m_1) \oplus z_m(m_2))))$ ;    /* Get top $N$ forward templates */
**for** $i \leftarrow 1$ *to* $N$ **do**
    **if** $t'[i]$ *is unimolecular* **then**
        $p \leftarrow t'[i](m_1)$ ;                      /* Apply fwd. template to $m$ */
        $G_F.\text{ADD\_RXN}(\{m_1\}, p, t'[i])$ ;    /* Add reaction and product to $G_F$ */
    **else**                                  /* $t'[i]$ is bimolecular */
        $b \leftarrow \text{KNN}_\mathcal{B}(\text{MLP}_b(z_m(m_1) \oplus z_m(m_2) \oplus z_t(t'[i])))$ ;  /* Get $K$ nearest BBs by cosine sim. */
        $\forall j \leftarrow 1$ to $K$: $G_F.\text{ADD\_RXN}(\{m_1, b\}, t'[i](m_1, b[j]), t'[i])$ ;    /* Apply $t'[i]$ */
    **end**
**end**

---

### 3.4 Extracting multi-step reaction data from a reaction corpus for offline learning

To learn $f_t$, $f_b$, and $D$, we approximate $\mathcal{G}$ by generating the incomplete network from a reaction dataset. In this work, we use the public USPTO-Full dataset [53, 54] of approximately 1 million deduplicated reactions. The dataset is filtered and processed (details in Section A.3), and a template set $\mathcal{T}_{\text{USPTO}}$ is extracted with RDChiral [55]. The dataset is randomly divided into training and validation splits with ratio 90:10. From the training split $\mathcal{R}_{\text{USPTO}}$ we construct the graph $\mathcal{G}_{\text{USPTO}}$. We filter reactions that **(1)** involve more than 2 reactants or **(2)** whose product cannot be recovered by applying the forward template $t'$, yielding $\mathcal{R}_{\text{FWD}}$, $\mathcal{G}_{\text{FWD}}$, and $\mathcal{T}'_{\text{FWD}}$.

Table 1: Benchmark dataset summary. Avg. In-Dist. % is the mean percentage of reactions in each route within the top 50 suggestions of the retro model. Unique Rxn.% is the ratio of deduplicated reactions to total reactions across all routes. Avg. # Rxns. is the mean number of reactions in each route, and Avg. Depth is the mean number of reactions in the longest path of each route.

| Dataset | # Routes | Avg. In-Dist. % | Unique Rxn. % | Avg. # Rxns. | Avg. Depth |
|---|---|---|---|---|---|
| **USPTO-190** | 190 | 65.6 | 50.5 | 6.7 | 6.0 |
| **Pistachio Reachable** | 150 | 100 | 86.1 | 5.5 | 5.4 |
| **Pistachio Hard** | 100 | 59.9 | 95.2 | 7.5 | 7.2 |

To learn $f_t$ and $f_b$, a full enumeration of all pathways (until reaching nodes in $\mathcal{B}$) rooted at $p^*$ is performed for each molecule node $p^*$ in $\mathcal{G}_{\text{FWD}}$. Reaction nodes in the enumerated pathways then each provide a training example for $f_t$ and $f_b$. Likewise, we enumerate pathways in $\mathcal{G}_{\text{USPTO}}$, and each molecule node $m$ in a pathway yields a training example for learning $D(m, p^*)$. The details for our training procedures are described in Section A.4.

Notably, we inject "negative" examples into our training set for $D$, as the distribution of costs is highly skewed towards low values. We define a modified MSE loss function as in Kim et al. [56] for learning $D$:

$$\mathcal{L} = \begin{cases} (y_{pred} - y_{true})^2 & \text{if } y_{true} \leq D_{max} \\ \max(0, D_{max} + 1 - y_{pred})^2 & \text{else} \end{cases} \tag{9}$$

This strategy allows the model to default to an approximate value of $D_{max} + 1$ for any "highly distant" molecule inputs. Now, for each target $p^*$, we randomly sample a molecule $m \in G_{\text{USPTO}}$ with no path to $p^*$ and Tanimoto similarity $< 0.7$ and add a training example with label $\infty$. In this work, we use $D_{max} = 9$.

# 4 Experiments

Our experiments are designed to answer the following: **(1)** Does DESP significantly improve the performance of starting material-constrained synthesis planning compared to baseline methods? **(2)** To what extent do $D$ and bidirectional search account for the performance of DESP? **(3)** Can DESP find routes to more complex targets than baseline methods? **(4)** What empirical differences do we see between F2E and F2F strategies?

## 4.1 Experimental setup

**Datasets** Few public datasets of multi-step synthetic routes exist. Previous works in multi-step synthesis planning have widely used the USPTO-190 dataset [6], a set of 190 targets with corresponding routes extracted from the USPTO-Full dataset. Others have tested on targets sampled from databases such as ChEMBL or GDB17 [57, 27, 31], but their lack of ground truth routes precludes the systematic selection of starting materials for our task. PaRoutes [58] has been proposed as an evaluation set, but they do not provide a standardized training set to prevent data leakage.

In addition to **USPTO-190**, because of its large proportion of out-of-distribution and redundant reactions (Table 1), we create and release two additional benchmark sets, which we call **Pistachio Reachable** and **Pistachio Hard**. Details of their construction are provided in Section A.6. To obtain the ground-truth goal molecules for each of our test sets, we find the longest path from target to leaf node in each route DAG and pick the leaf node with more heavy atoms. For the building block set $\mathcal{B}$, we canonicalize all SMILES strings in the set of 23 million purchasable building blocks from eMolecules used by Chen et al. [6].

**Model training** As in [6], we train a single-step retrosynthesis MLP (NeuralSym) and Retro* cost network on our processed training split of USPTO-Full. The synthetic distance and forward expansion models are trained as described in Sections 3.4 and A.4.

**Multi-step algorithms** Because we utilize an AND-OR search graph with no duplicate molecule nodes, our implementation of Retro* is more comparable to RetroGraph [30], but we do not employ

Table 2: Summary of starting material-constrained planning performance across the three benchmarks. Solve rate refers to the percentage of $(p^*, r^*)$ pairs in the dataset solved at the given expansion limits. The average number of expansions $\overline{N}$ is given for each method, with a max budget of 500.

| Algorithm | USPTO-190 | | | | Pistachio Reachable | | | | Pistachio Hard | | | |
|---|---|---|---|---|---|---|---|---|---|---|---|---|
| | Solve Rate (%) ↑ | | | $\overline{N}$ ↓ | Solve Rate (%) ↑ | | | $\overline{N}$ ↓ | Solve Rate (%) ↑ | | | $\overline{N}$ ↓ |
| | $N$=100 | 300 | 500 | | 50 | 100 | 300 | | 100 | 300 | 500 | |
| Random | 4.2 | 4.7 | 4.7 | 479 | 16.0 | 26.7 | 40.7 | 325 | 6.0 | 12.0 | 13.0 | 452 |
| BFS | 12.1 | 20.0 | 24.2 | 413 | 48.7 | 57.3 | 74.0 | 169 | 16.0 | 26.0 | 29.0 | 390 |
| MCTS | 20.5 | 32.1 | 35.3 | 364 | 52.0 | 72.7 | 85.3 | 111 | 27.0 | 31.0 | 32.0 | 361 |
| Retro* | 25.8 | 33.2 | 35.8 | 351 | 70.7 | 78.0 | 92.7 | 73 | 32.0 | 35.0 | 37.0 | 342 |
| GRASP | 15.3 | 21.1 | 23.7 | 410 | 46.7 | 51.3 | 66.7 | 198 | 14.0 | 22.0 | 29.0 | 402 |
| Bi-BFS | 20.0 | 26.3 | 28.4 | 382 | 66.0 | 75.3 | 86.0 | 101 | 28.0 | 34.0 | 38.0 | 341 |
| Retro*+$D$ | 27.4 | 32.6 | 37.4 | 348 | 77.3 | 87.3 | *96.0* | 49 | 31.0 | 40.0 | 42.0 | 323 |
| DESP-F2E | **30.0** | **35.3** | **39.5** | *340* | *84.0* | **90.0** | *96.0* | *41* | *35.0* | *44.0* | **50.0** | *300* |
| DESP-F2F | *29.5* | *34.2* | **39.5** | **336** | **84.5** | *88.9* | **97.3** | **38** | **39.0** | **45.0** | *48.0* | **293** |

their GNN guided value estimation and thus refer to the algorithm as Retro* for simplicity. This serves as both a baseline and ablated version of DESP (without bidirectional search or $D$). In addition, we test the performance of random selection, breadth-first search (BFS), bidirectional-BFS, and MCTS. Finally, we integrate our single-step model into GRASP [13] using the authors' published code. Since their data is not publicly available, we retrained their model on 10,000 randomly sampled targets in our training set and run their search implementation on each benchmark. For all methods, we enforce a maximum molecule depth of 11, a maximum of 500 total expansions (retro or forward), and apply 50 retro templates per expansion. For DESP, we also enforce a maximum molecule depth of 6 for the bottom-up search, apply 25 forward templates per expansion, and use the top 2 building blocks found in the k-NN search. Due to the asymmetry of the bidirectional search, we also introduce a hyperparameter $\lambda$, the number of times we repeat a select, expand, and update cycle for $G_R$ before performing one cycle for $G_F$. For all experiments, we set $\lambda = 2$. Details and tabular summaries of the evaluations performed and hyperparameters chosen are provided in Section A.7.

## 4.2 Results

Though it is notoriously difficult to quantitatively evaluate synthetic routes proposed *in silico* without expert evaluation, there are widely-used metrics thought to correlate with successful algorithms, such as higher solve rate (under varying computational budgets), lower average number of expansions, and lower average number of reactions in found routes [59, 57]. We focus on these metrics, as they are arguably most related to a search algorithm's efficiency. Because all methods employ the same one-step model and set of templates from USPTO-Full, we treat their proposals as equally chemically feasible.

Table 3: Average $\pm$ stdev of the number reactions in proposed routes. Comparisons are made across $(p^*, r^*)$ pairs solved by all methods.

| Dataset | USPTO-190 | Pistachio Easy | Pistachio Hard |
|---|---|---|---|
| # Routes | 63 | 139 | 36 |
| | Avg. # Rxns. ↓ | | |
| Retro* | **5.56** $\pm$ 2.37 | 4.94 $\pm$ 2.27 | 4.81 $\pm$ 2.09 |
| Retro*+$D$ | 5.87 $\pm$ 2.37 | *4.92* $\pm$ 2.27 | *4.80* $\pm$ 2.08 |
| DESP-F2E | **5.56** $\pm$ 2.55 | **4.86** $\pm$ 2.17 | **4.67** $\pm$ 2.35 |
| DESP-F2F | 5.95 $\pm$ 3.93 | 5.17 $\pm$ 2.37 | *4.78* $\pm$ 2.60 |

**Improvement on starting material-constrained synthesis planning**  Quantitative benchmarking results are summarized in Table 2. Both variants of DESP outperform all baseline methods in terms of solve rate and average number of expansions across all test sets. The solve rates of baseline methods on USPTO-190 are noticeably lower than commonly reported ranges for general synthesis planning [6], as the starting material constraint increases the difficulty of the task. Notably, unlike the other benchmarked methods, the Random, BFS, MCTS, and Retro* are standard single-ended search methods that do not make any use of the starting material constraint information.

**Ablation studies**  To investigate the contributions of $D$ and bidirectional search, we conduct an ablation study by running Retro* guided by $D$ on all benchmarks (denoted as Retro*+$D$ in Tables 2 and 3). We find that incorporating $D$ generally results in higher solve rates and fewer average

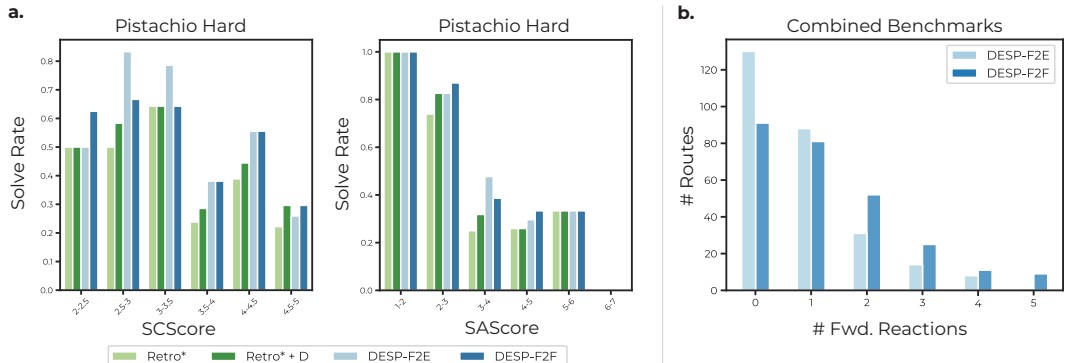

Figure 3: Ablation study. **(a)** Solve rate as a function of the binned complexity of target molecules in Pistachio Hard. **(b)** Number of forward reactions in `DESP` routes across all benchmark sets.

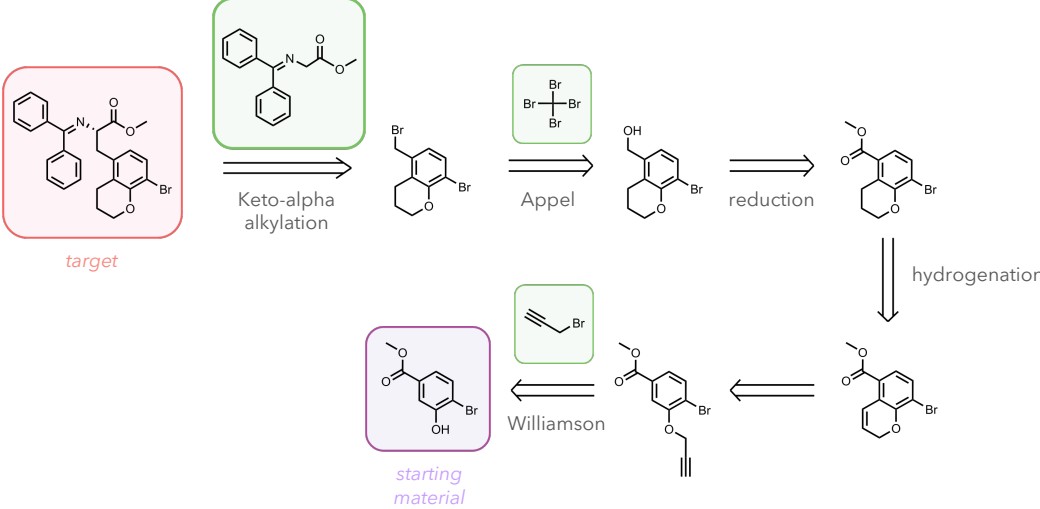

Figure 4: Exemplary synthetic route for a test case that `DESP-F2F` was able to solve but Retro* was unable to solve. `DESP-F2F` was able to match every step of the reference route in this case.

expansions across all datasets, but still does not outperform `DESP`, demonstrating the role of both $D$ and bidirectional search in improving planning efficiency. As an indicator of route quality, we also investigate the average number of reactions in the outputs of `DESP` (Table 3). `DESP-F2E` is able to find shorter routes on average when compared to either Retro* or Retro* guided by $D$. An example of a route solved by `DESP-F2F` (but not by Retro*) is visualized in Fig. 4.

**Performance on complex targets**  To investigate the degree to which `DESP` improves planning performance on complex targets, we bin each target in Pistachio Hard by two commonly-used metrics of synthetic complexity, SCScore [60] and SAScore [61]. Both variants of `DESP` equal or outperform Retro* on solve rates across all complexity ranges (Fig. 3a). This indicates that, in the starting material-constrained setting, `DESP` can improve planning performance on targets of higher complexity, a regime which current CASP algorithms struggle with.

**F2E and F2F comparisons**  Though `DESP-F2F` consistently expands slightly fewer nodes on average, the empirical differences in efficiency between F2E and F2F are small. However, `DESP-F2E` is able to find noticeably shorter routes on average than `DESP-F2F`, which finds routes even longer than Retro* on multiple benchmarks (Table 3). A likely reason for this difference is due to the lack of consideration of the pathway depth of existing nodes in front-to-front search, which we elaborate on in Section A.8. We also investigate the extent to which reactions from forward expansions end up in the solutions of each variant. As visualized in Fig. 3b, `DESP-F2F` incorporates more forward reactions, while `DESP-F2E` solutions are dominated by top-down search almost half the time. We hypothesize

that the difficulty of bottom-up synthesis planning [42] further contributes to the increased length of `DESP-F2F` solutions, as `DESP-F2F` empirically relies more on forward reactions and thus is likely more sensitive to the performance of the forward models.

## 5  Conclusion

In this work, we introduce `DESP`, a novel framework for bidirectional search as applied to computer-aided synthesis planning. `DESP` biases searches towards user-specified starting materials with a combination of a learned synthetic distance network and bottom-up generation of part of the synthetic route. This represents a task that aligns with a common use case in complex molecule synthesis planning. We demonstrate the efficiency of `DESP` on the USPTO-190 dataset and two new test sets derived from the Pistachio database. When compared to existing methods, both variants of `DESP` reduce the number of expansions required to find solutions that satisfy the specified goal, while `DESP-F2E` also finds more routes with fewer reactions per route. We anticipate that with future improvements to the synthetic distance network and bottom-up synthesis planning, bidirectional synthesis planning can emerge as an effective and efficient solution to navigating constraints and aligning computer-aided synthesis planning with the goals of domain experts. Additional outlook is provided in Section A.8.

## Code and Data Availablity

Relevant code with documentation can be found at `https://github.com/coleygroup/desp`.

## Acknowledgments and Disclosure of Funding

This research was supported by the Machine Learning for Pharmaceutical Discovery and Synthesis consortium. J.R. acknowledges funding support from the NSF Center for Computer Assisted Synthesis (C-CAS) under Grant CHE-2202693. W.G. is supported by the Google Ph.D. Fellowship and Office of Naval Research under grant number N00014-21-1-2195. We thank Prof. Sarah Reisman and Prof. Richmond Sarpong for discussions during the ideation of the project. We thank Dr. Roger Sayle and NextMove Software for granting us permission to release the Pistachio-derived benchmark sets. We thank Prof. Yunan Luo for providing computational resources used in an earlier prototype of `DESP`. We thank Dr. Babak Mahjour for discussions around the chemical feasibility of proposed routes.

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

# A  Appendix / supplemental material

## A.1  Summary of notation

| Symbol | Note |
|--------|------|
| $\mathcal{M}$ | set of all valid molecules |
| $\mathcal{R}$ | set of all valid reactions |
| $\mathcal{T}$ | set of all valid retro templates |
| $\mathcal{T}'$ | set of all valid single-product fwd. templates |
| $\mathcal{B}$ | set of building blocks, where $\mathcal{B} \subset \mathcal{M}$ |
| $\mathcal{G}$ | the AND-OR graph constructed from all possible reaction tuples $\mathcal{R}$ |
| $R$ | single-product reaction |
| $t$ | retro reaction template |
| $t'$ | fwd. reaction template |
| $p^*$ | target molecule |
| $r^*$ | starting material goal |
| $S$ | valid synthetic route |
| $c$ | reaction cost function |
| $\gamma(m)$ | given $m$, opposing molecule to condition selection or expansion on |
| $G_R$ | top-down search graph |
| $G_F$ | bottom-up search graph |
| $\mathcal{F}_R$ | frontier molecule nodes in $G_R$ |
| $\mathcal{F}_F$ | frontier molecule nodes in $G_F$ |
| $V_m$ (retro) | estimated minimum cost of synthesizing $m$ |
| $V_m$ (fwd.) | estimated value of $D(m, \gamma(m))$ |
| $rn(m|G)$ | "reaction number," estimated cost of synthesizing $m$ given search graph $G$ |
| $V_t(m|G)$ | estimated cost of synthesizing $p^*$ using $m$ given search graph $G$ |
| $D$ | synthetic distance (network) |
| $D_m$ | estimated value of $D(\gamma(m), m)$ |
| $dn(m|G)$ | "distance numbers," multiset of descendent $D_m - V_m$ values for $m$ in $G$ |
| $D_t(m|G)$ | multiset of related $D_m - V_m$ values for $m$ in $G$ |
| $f_t$ | forward template predictor model |
| $f_b$ | building block predictor model |
| $\mathcal{L}$ | loss function |
| $D_{max}$ | maximum value of $D$ considered in $\mathcal{L}$ |
| $\lambda$ | # retro expansions to perform before one fwd. expansion |

## A.2  Retro* algorithm details

Retro* defines the following quantities:

1. $V_m$. For a molecule $m$, $V_m$ is an unconditional estimate of the minimum cost required to synthesize $m$. It is estimated by a neural network.

2. $rn(m|G)$. For a molecule $m$, given search graph $G$, the "reaction number" $rn(m|G)$ represents the estimated minimum cost of synthesizing $m$.

3. $V_t(m|G)$. For a molecule $m$, given search graph $G$ with goal $p^*$, $V_t(m|G)$ represents the estimated minimum cost of synthesizing $p^*$ using $m$.

Retro* also cycles between selection, expansion, and update phases. We implement Retro* as follows.

**Selection**  The molecule in the set of frontier nodes $\mathcal{F}$ that minimizes the expected cost of synthesizing $p^*$ given the current search graph $G$ is selected:

$$m_{select} = \arg\min_{m \in \mathcal{F}} V_t(m|G) \tag{10}$$

**Expansion**  As in Alg. 2, a one-step retrosynthesis model is called on the selected node and the resulting reactions and precursors are added to $G$. Each molecule node is then initialized with $rn(m|G) \leftarrow V_m$.

**Update** First, reaction number values are uppropagated to ancestor nodes. For reaction node $R$, the reaction number is updated as the sum of its childrens' reaction numbers.

$$rn(R|G) \leftarrow c(R) + \sum_{m \in ch(R)} rn(m|G) \tag{11}$$

For molecule node $m$, the reaction number becomes the minimum reaction number among its children.

$$rn(m|G) \leftarrow \min_{R \in ch(m)} rn(R|G) \tag{12}$$

Next, $V_t(m|p^*)$ values are downpropagated to descendent nodes. Starting from $p^*$ itself,

$$V_t(p^*|G) \leftarrow rn(p^*|G) \tag{13}$$

For subsequent reaction nodes $R$, the value is updated

$$V_t(R|G) \leftarrow rn(R|G) - rn(pr(R)|G) + V_t(pr(R)|G) \tag{14}$$

Finally, for molecule node $m$ that is not $p^*$,

$$V_t(m|G) \leftarrow \min_{R \in pr(m)} rn(R|G) \tag{15}$$

### A.3 Reaction pre-processing

Reactions in the USPTO-Full dataset are represented with simplified molecular-input line-entry system (SMILES) [62] strings, where the SMILES string of reactants, reagents, and products are separated by '>' as `REACTANTS>REAGENTS>PRODUCTS`. Each field can have one or more chemical species delineated with a dot (.) or be left blank in the case of reagents.

For processing reaction SMILES, multi-product reaction SMILES are first separated into single-product reaction SMILES by creating separate entries for each product species with the same reactants and reagents. Each single-product reaction SMILES then undergoes the following process:

1. Reagents in the SMILES string are moved to the reactant side.
2. Chemical species with identical atom mapped SMILES in both reactants and products are moved to reagents.
3. Any products that do not contain at least one mapped atom or have fewer than 5 heavy atoms are removed.
4. Any atom mapping numbers that exist exclusively on either the reactant side or product side are removed.
5. Any reactants without atom mapping are moved to the reagent side.

Resulting reaction SMILES without either reactants or products are then filtered out.

### A.4 Model training details

**Dataset construction** To learn $f_t$ and $f_b$, a full enumeration of all pathways (until reaching nodes in $\mathcal{B}$) rooted at $p^*$ is performed for each molecule node $p^*$ in $\mathcal{G}_{\text{FWD}}$. For learning $f_t$, each reaction node $R_i = (r_i, p_i, t_i)$ is then used as a training example for each reactant $m_j \in r_i$ with input $z_m^{n_1}(m_j) \oplus z_m^{n_1}(p^*)$ and one-hot encoded label $t_i$. Likewise, for learning $f_b$, each reaction node $R_i = (\{m_1, m_2\}, p_i, t_i)$ that is bimolecular and involves at least one building block yields a training example with input $z_m^{n_1}(m_1) \oplus z_m^{n_1}(p^*) \oplus z_t^{n_1}(t_i)$ and output $z_m^{n_2}(m_2)$ if $m_2 \in \mathcal{B}$ and with input $z_m^{n_1}(m_2) \oplus z_m^{n_1}(p^*) \oplus z_t^{n_1}(t_i)$ and output $z_m^{n_2}(m_1)$ if $m_1 \in \mathcal{B}$. The procedure for such training example generation is illustrated in Fig. 5. With $n_1 = 2048, n_2 = 256$, we use the RDKit implementation of the Morgan Fingerprint [63] with radius 2 for $z_m$ and the Atom Pair fingerprint [64] for $z_t$.

Because $D$ is used to bias both the top-down and bottom-up searches, we perform the same pathway enumeration for all molecule nodes $p^* \notin \mathcal{B}, p^* \in \mathcal{G}_{\text{USPTO}}$. In this case, however, we only consider $p^*$ for which we find valid synthetic routes. Training examples are then extracted for all other molecule nodes $m_i$ in a solved search graph, with input $z_m^n(m_i) \oplus z_m^n(p^*)$ and label $V_{p^*}(m_i|G_R) - rn(m_i|G_R)$, with $n = 512$. For calculating this label, we propagate the Retro* functions as described in Section

A.2 such that $D$ can be calculated as the minimum cost of synthesizing $p^*$ subtracted by the minimum cost of synthesizing $m_i$. Here, we set $c(R_i) = 1$ for all $R_i$, as a synthetic route's number of steps is an important metric in evaluating the route cost, and it is otherwise difficult to objectively quantify the cost of a reaction. This training example generation is also depicted in Fig. 5. Finally, to obtain additional training examples, we also recover pairs of $(m, p^*)$ where $p^*$ was not "solved" by the enumerative search but would have been solved if $m \in \mathcal{B}$.

For validation of the $f_t$ and $f_b$ models, we construct the graph $\mathcal{G}_{\text{val}}$ corresponding to all reactions across both the training and validation splits. We perform the same pathway enumeration described above, and each "training example" that corresponds to a reaction not in the original training split is used as a validation example. For validation of the $D$ model, we construct $\mathcal{G}_{\text{val}}$ from $\mathcal{R}_{\text{USPTO}}$ and perform the pathway enumeration only on $p^* \notin \mathcal{G}_{\text{USPTO}}$ to obtain validation examples.

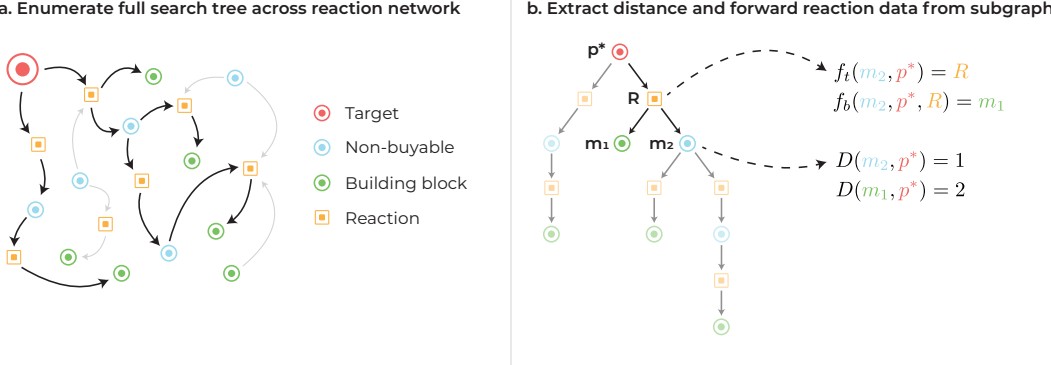

Figure 5: Illustration of data extraction procedure for offline training of $f_t$, $f_b$, and $D$. **(a)** For each target, the full search graph is enumerated by recursively tracing outgoing edges and propagating Retro* quantities. **(b)** For each bimolecular reaction with at least one buyable reactant, training examples for $f_t$ and $f_b$ are labeled. For each molecule node $m$ other than the target, $D(m, p^*)$ is computed.

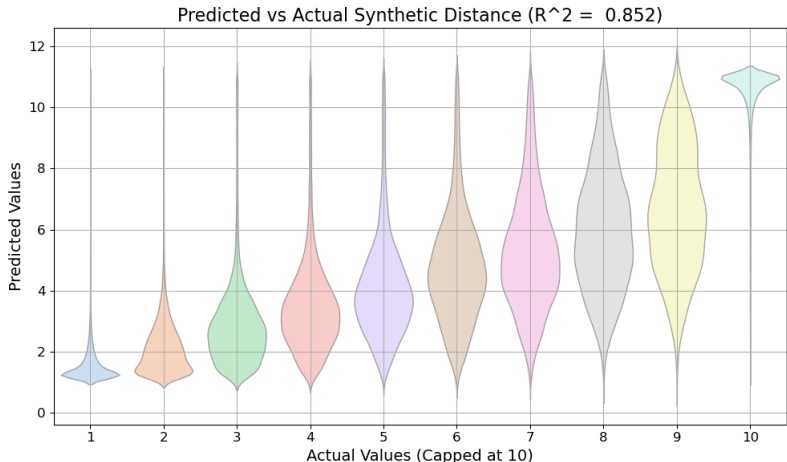

Figure 6: Predicted vs. actual values of synthetic distance on the validation examples. Actual values above 9 are set to 10.

**Model hyperparameters**   All models are MLPs trained with the Adam optimizer, early stopping (patience 2), and decayed learning rate on plateau with factor 0.3 and patience 1 on a single NVIDIA RTX 4090. The following table summarizes the hyperparameters and details of each model used in experiments.

| Model | Batch Size | Dropout | Activation | # Hidden Layers | Hidden Units | Learning Rate | Input Dim. | Output Dim. |
|-------|-----------|---------|-----------|-----------------|--------------|---------------|-----------|-------------|
| Retro Template | 2048 | 0.5 | Sigmoid | 2 | 2048 | 0.002758 | 2048 | 270794 |
| Fwd Template | 4096 | 0.4 | SiLU | 2 | 1024 | 0.005113 | 4096 | 196339 |
| BB Model | 4096 | 0.3 | ReLU | 3 | 2048 | 0.001551 | 6144 | 256 |
| Retro* $V_m$ | 4096 | 0.2 | SiLU | 4 | 128 | 0.0025 | 2048 | 1 |
| Synthetic Dist. $D$ | 4096 | 0.3 | Sigmoid | 4 | 256 | 0.00489 | 1024 | 1 |

Hyperparameters were selected based on best performance on the validation split while performing a Bayesian search through the following parameter space:

1. Dropout: [0.1, 0.2, 0.3, 0.4, 0.5]

2. Activation: [ReLU, SiLU, Sigmoid, Tanh]

3. Hidden layers: [2, 3, 4]

4. Hidden units: [1024, 2048] for retro, forward, and BB. [128, 256] for $D$ and $V_m$

5. Learning rate: [0.00001 - 0.01]

The template relevance module from the open-source ASKCOS codebase was used to train the one-step retro model.[1]

### A.5  DESP **additional details**

**a. Uppropagation**

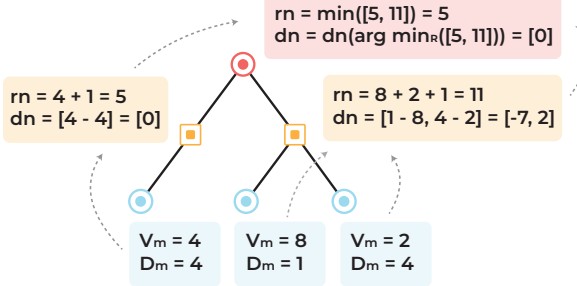

**b. Downpropagation**

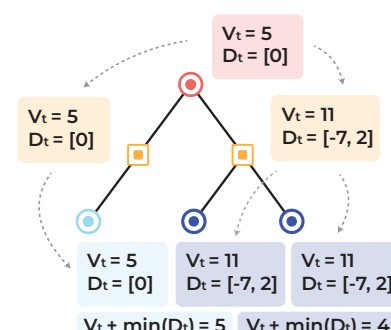

Figure 7: Simple visual example of DESP update procedure for guiding top-down search with synthetic distance, where each reaction has cost 1. In the unguided Retro* algorithm, the left-most frontier node would be chosen for expansion, as a route through that node minimizes $V_t(m|G)$, the expected cost to the target from building blocks (5 reaction steps). In DESP, either of the other two nodes would be prioritized, as the middle frontier node is predicted to be only 1 reaction step away from the starting material, resulting in only $V_t(m|G) + \min D_t(m|G) = 4$ predicted reaction steps total in the final solution.

---

**Algorithm 2:** RETRO_EXPAND($m$, $G_R$, $N$)

$m$: expanded molecule node, $G_R$: top search graph, $N$: num. templates to propose

---

$t \leftarrow \text{TOP\_N}(\sigma(\text{MLP}_R(z_m(m))))$ ;   /* Get top $N$ templates from retro model */
**for** $i \leftarrow 1$ *to* $N$ **do**
   $r \leftarrow t[i](m)$ ;           /* Apply retro template to $m$ */
   $G_R.\text{ADD\_RXN}(r, m, t[i])$;   /* Add reaction and precursors to $G_R$ */
**end**

---

[1]Template relevance module can be found at `https://gitlab.com/mlpds_mit/askcosv2/retro/template_relevance`.

**Algorithm 3:** DESP($p^*$, $r^*$, $N_1$, $N_2$, $K$, $L$, $\lambda$, $s$)

$p^*$: target, $r^*$: starting material goal, $N_1$: num. retro templates to propose, $N_2$: num. forward templates to propose, $K$: num. building blocks to search, $L$: max num. expansions, $\lambda$: num. times to expand top before expanding bottom, $s$: strategy (F2E or F2F)

---

$G_R \leftarrow \{p^*\}$ ;                                           /* Initialize search graphs */
$G_F \leftarrow \{r^*\}$;
$l \leftarrow 0$;
**while** *not solved OR* $l \leq L$ **do**
    **for** $i \leftarrow 1$ *to* $\lambda$ **do**
        $m \leftarrow \arg\min_{m \in \mathcal{F}_R} [V_t(m|G_R) + \min(D_t(m|G_R))]$ ;     /* Select best top */
        RETRO_EXPAND($m, G_R, N_1$) ;                              /* Expand with Alg.2 */
        TOP_UPDATE($G_R$);                         /* Update $G_R$ with Section 3 rules */
        **if** *met bottom* **then**
            $m_{met}.rn \leftarrow 0$ ;               /* Set expected cost of met node to 0 */
            BOT_UPDATE($G_F$);                          /* Retro* updates on $G_F$ */
        **end**
        $l \leftarrow l + 1$;
    **end**
    $m \leftarrow \arg\min_{m \in \mathcal{F}_F} V_t(m|G_F)$ ;                          /* Select best bot */
    **if** $s = F2E$ **then**
        FORWARD_EXPAND($m, p^*, G_F, N_2, K$);     /* Expand conditioned on $p^*$ */
        BOT_UPDATE($G_F, s$);                          /* Retro* updates $G_F$ */
    **else if** $s = F2F$ **then**
        $q \leftarrow \arg\min_{q \in G_R} D(m, q)$;                     /* Find closest node */
        FORWARD_EXPAND($m, q, G_F, N_2, K$);     /* Expand conditioned on $q$ */
        BOT_UPDATE($G_F, s$);                          /* Retro* updates on $G_F$ */
    **if** *met top* **then**
        $m_{met}.rn \leftarrow 0, m_{met}.dn \leftarrow [0]$ ; /* Set expected costs of met node to 0 */
        TOP_UPDATE($G_F$);                          /* Section 3 updates on $G_R$ */
    **end**
    $l \leftarrow l + 1$;
**end**

---

**Design of new quantities and update rules** We recall that the minimum total cost of synthesizing the target $p^*$ from a molecule $m$ under the Retro* framework is estimated as:

$$V_t(m|G_R) = \sum_{r \in \mathcal{A}^r(m|G_R)} c(r) + \sum_{m' \in \mathcal{V}^m(G_R), pr(m') \in \mathcal{A}^r(m|G_R)} rn(m'|G_R) \tag{16}$$

where $\mathcal{A}(m|G_R)$ represents the set of reaction node ancestors of $m$ and $\mathcal{V}^m(G_R)$ represents the set of molecule nodes in the search graph. This is equivalent to

$$V_t(m|G_R) = g(m|G_R) + \sum_{m' \in \mathcal{N}(m|G_R)} V_{m'} \tag{17}$$

where $g(m|G_R)$ aggregates the current cost from all reaction nodes in $G_R$ contributing to $V_t(m|G_R)$, and $\mathcal{N}(m|G_R) \subseteq \mathcal{F}_R$ accordingly represents the set of frontier top nodes for the subgraph of $G_R$ corresponding to nodes contributing to $V_t(m|G_R)$. If we add the constraint that one frontier node must implicitly be the ancestor of $r^*$, the estimate of the minimal cost then becomes:

$$V'_t(m|G_R) = g(m|G_R) + \min_{m_j \in \mathcal{N}(m|G_R)} \left( \sum_{m_i \in \mathcal{N}(m|G_R), m_i != m_j} V_{m_i} + D(r^*, m_j) \right) \tag{18}$$

$$= g(m|G_R) + \sum_{m_i \in \mathcal{N}(m|G_R)} V_{m_i} + \min_{m_j \in \mathcal{N}(m|G_R)} \left( D(r^*, m_j) - V_{m_j} \right) \tag{19}$$

$$= V_t(m|G_R) + \min_{m_j \in \mathcal{N}(m|G_R)} D_{m_j} \tag{20}$$

Our update rules are implemented such that $D_t(m|G_R) = \min_{m_j \in \mathcal{N}(m|G_R)} D_{m_j}$, thus justifying our design of the selection and update procedures. Note that this design relies on the assumption that $\mathcal{N}(m|G_R)$ remains static upon adding the goal node constraint, when in reality the introduction of $D$ may change the optimal set of frontier nodes to consider in the search graph. To avoid the combinatorial complexity of this situation and retain the efficiency from dynamic programming for our update policy, we maintain this assumption and find that introducing $D$ in this way empirically works well (Section 4.2). A simple visual example of the update procedures is provided in Figure 7.

### A.6 New benchmark set details

We follow the test set extraction procedure of Chen et al. [6], applied within patents of the Pistachio dataset [14] (version: 2023Q4) to obtain 1,004,092 valid synthetic routes. We randomly sample synthetic routes from this set until we obtained 150 routes that satisfied the following constraints: (1) No reactions in the route are found in the training data. (2) No reactions are shared between any routes within the test set. (3) All reactions are found in the top 50 proposals of our single-step retrosynthesis model. (4) No two targets in the test set have a Tanimoto similarity of more than 0.7. (5) We enforce a minimum number of routes for different route lengths (Fig. 8, Fig. 9). We term this set of 150 routes **Pistachio Reachable**. We perform the same procedure but modify condition (2) to require only 50% or more of the reactions to be reproducible (in-distribution) and obtain 100 routes which we term **Pistachio Hard**. Due to a bug in our implementation of criterion (2), a small number of routes share the same reaction in the final datasets, but the degree of inter-route reaction duplication is still significantly less than that of USPTO-190 for both benchmark sets (Table 1).

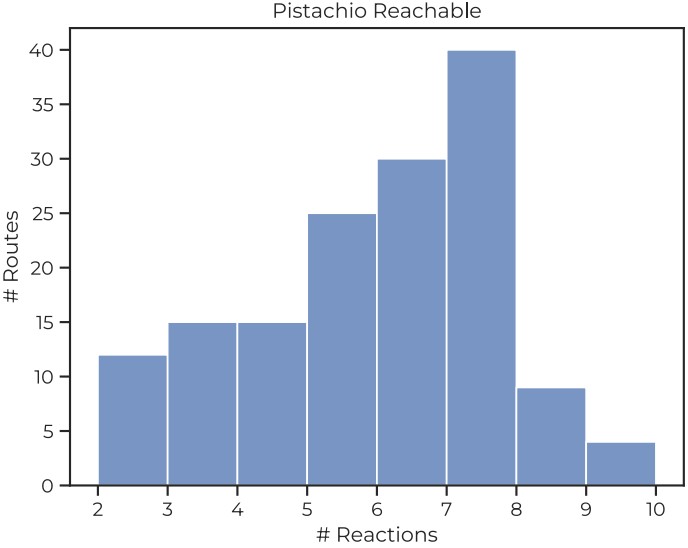

Figure 8: Distribution of reaction counts in Pistachio Reachable.

### A.7 Additional experimental details

**Implementation details** For **random search**, all node selections were performed at random among frontier molecule nodes. For **BFS**, the molecule with the lowest depth was selected at each step, with precedence for nodes whose parent reactions had the highest plausibility scores from the retro one-step model. **MCTS** was run by integrating our one-step model into the open-source ASKCOS code base [65]. For **Retro\***, we removed the synthetic distance network and bottom-up expansions from our DESP implementation. Notably, reaction costs for Retro* and DESP are both calculated as $-\log p$, where $p$ is the template plausibility from the one-step model (retro or forward). For **GRASP**, we used the authors' search implementation [13]. For a fair comparison with the AND-OR graph structure, we did not increment the iteration counter when a molecule that had previously been expanded was expanded again. In training the GRASP value network, we use the authors' reported hyperparameters where applicable and the default values in their code base otherwise.

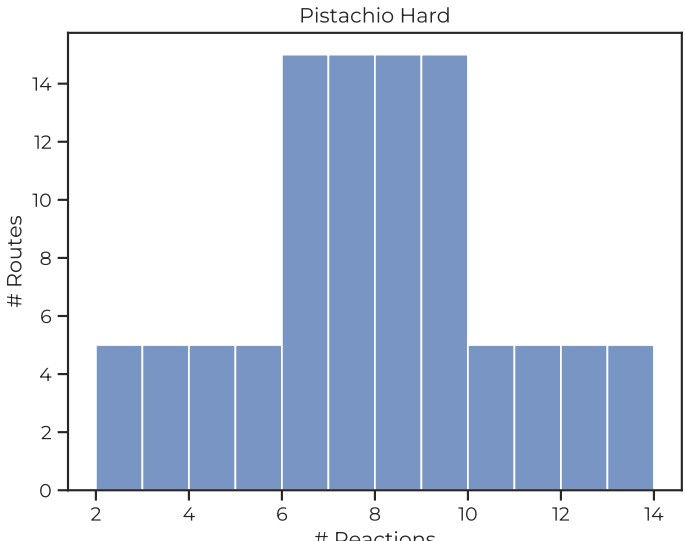

Figure 9: Distribution of reaction counts in Pistachio Hard.

Table 4: Summary of hyperparameters used for evaluated algorithms.

| Algorithm | Hyperparameter | Value |
|---|---|---|
| All | Max # expansions | 500 |
| | Max mol. depth (top) | 11 |
| | Max mol. depth (bottom) | 6 |
| | Max # retro templates | 50 |
| DESP / Bi-BFS | Max # fwd. templates | 25 |
| | Max # BBs retrieved | 2 |
| | $\lambda$ | 2 |
| MCTS | Exploration weight | 1.0 |

Table 5: Summary of components of each evaluated algorithm.

| Algorithm | Selection Policy | Guidance | Bidirectional? |
|---|---|---|---|
| Random | Randomly selected frontier node | None | No |
| BFS | Lowest depth frontier node, ties broken by reaction cost | None | No |
| MCTS | Start from root and use UCB to select children until reaching frontier node | None | No |
| Retro* | Node minimizing $V_t(m\|G)$ | BB | No |
| GRASP | Same as MCTS | BB *or* s.m. | No |
| Bi-BFS | Same as BFS | None | Yes |
| Retro* + $D$ | Node minimizing $V_t(m\|G) + \min(D_t(m\|G))$ | BB *and* s.m. | No |
| DESP | Alternate between top-down and bottom-up, both using Retro* and $D$ | BB *and* s.m. *and* target | Yes |

**Approximate nearest neighbors search** In selecting building blocks for the forward expansion with k-NN search, the Python library `Faiss` is used. A 256-dimension Morgan fingerprint of each building block is stored in a vector database and compressed using product quantization for approximate nearest neighbor search at dramatically faster speeds and significantly lower memory overhead.

**Compute** All experiments were performed on a 32-core AMD Threadripper Pro 5975WX processor and with a single NVIDIA RTX 4090 GPU. Running experiments on all benchmark sets for a given

method took around 10 hours. DESP requires $\sim 3$ GB of GPU memory to store the building block database for fast k-NN.

## A.8 Limitations and Outlook

**Convergent syntheses**    Convergent synthetic routes, in which multiple non-BBs are combined, are often desirable in chemistry due to their relative efficiency. The top-down search has no problems proposing convergent routes. However, the bottom-up searcher in DESP only performs forward expansions and thus cannot handle convergent routes by adding and merging new synthetic trees. Resultantly, the bottom-up search can only plan one branch if the final route requires convergent steps. Implementing additional modules of SynNet [42] into the bottom-up search would enable planning of convergent synthetic routes and potentially further reduce the average number of reactions in solutions and improve solve rates.

**GPU reliance and computational overhead**    DESP requires GPU acceleration to tractably perform a k-NN search over $\sim 23$ million building blocks in the forward expansion policy. DESP-F2F also requires GPU inference to rapidly perform node comparisons at each iteration. In all, forward expansions take around 50% more time than retro expansions, though this is in part because our implementation of forward synthesis applies retro templates to each product proposed by the forward model to ensure template reversibility (i.e., to confirm that the increased success in finding routes during the bidirectional search is not an artifact of having access to "different" transformations), which creates additional overhead. Overall, we view these limitations primarily as engineering problems that do not take away from the empirical benefits demonstrated in the paper. In principle, one could also implement DESP-F2E as a parallel bidirectional search in pursuit of additional efficiency gains.

**Building block specification**    Though DESP is designed to address starting material-constrained synthesis planning, we envision that future work could optimize bidirectional search to improve general retrosynthesis capabilities by *conditioning* on one or more starting materials instead of *constraining* the solution space. These starting materials could be expert-designed or predicted algorithmically as in Gao et al. [42].

DESP-F2F **implementation**    In general, neither DESP-F2E or DESP-F2F guarantee that the cost-optimal route is found upon termination. Additionally, our implementation of DESP-F2F does not take into account the total known cost of the opposing graph's nodes $V_t(m'|G_F) - rn(m'|G_F)$ when calculating $dn(m|G_R)$, and likewise the value of $rn(m|G_F)$ does not take into account $V_t(m'|G_R) - rn(m'|G_R)$. As a result, the selection policy DESP-F2F selects nodes that minimize the lowest expected cost of reaching the opposing search graph, but does not select to minimize the lowest expected cost of the final route directly. This is likely a primary contributor to DESP-F2F finding longer routes on average than DESP-F2E. As the values of $V_t(m|G)$ change after each expansion, it would be computationally expensive to re-compare nodes across the search graphs at each iteration. We have not devised an efficient means of handling the number of re-comparisons that would be required and leave such optimizations to future exploration.

**Sub-goal and divide-and-conquer strategies**    Goal-oriented synthesis planning bring to mind potential methods that involve sub-goals or divide-and-conquer strategies that have been utilized in general purpose planning [66] or program synthesis [67]. In general, there are rich bodies of literature in fields like hierarchical planning and program synthesis that remain largely untapped in applications to computer-aided synthesis planning.

