# OpenReview forum: "Double-Ended Synthesis Planning with Goal-Constrained Bidirectional Search"
_NeurIPS.cc/2024/Conference — NeurIPS 2024 spotlight_

### Official Review · Reviewer_FuFE · 2024-06-28

**Soundness:** 3
**Presentation:** 3
**Contribution:** 3
**Rating:** 7
**Confidence:** 3

**Summary:**

The paper addresses the synthesis planning problem where constrains are taken into account. To that end, the authors propose a double-ended synthesis planning grounded with bidirectional search to ensure that the added constrains are met. They show experimentally that their proposed approach helps with solve rate as well as reduction in number of node expansion.

**Strengths:**

- The paper presents a novel idea.
- The real-world aspect of having constraint is addressed in this paper
-  Experiments have a clear objective and look sound.

**Weaknesses:**

- See the questions section.
- Proof of soundness and completeness of algorithm 1 is missing. In particular, what can be said about the properties of this algorithm?
- This paper provides a novel contribution for synthesis planning, but I fail to see its broader impact to the community (its significance is of narrow scope).

**Questions:**

1. Why Bi-directional search? Have you tried other search algorithms? You can potentially still define goal constrains and be in the forward search algorithm and ensure those goals are met. Forward search algorithms with heuristic search can address soft and hard constraints on goals or even on the full trajectory. Some examples follow. While these examples are from Automated planning literature, perhaps some can be relevant in this context. Note the specification of constraints in Linear Temporal Logic, as well as heuristics to meet these constraints has been a topic of interest in several publications.
    1. https://www.cs.toronto.edu/~sheila/publications/bai-mci-aim08.pdf
    2. https://www.sciencedirect.com/science/article/pii/S0004370208001975
    3. https://cdn.aaai.org/ICAPS/2005/ICAPS05-019.pdf

2. Could there be multiple solutions to the general synthesis planning problem defined in Section 2.2. If so is the notion of cost you mentioned taken into account? Also is the algorithm in section 3.2 going to favor the least costly solution? If so do you have a proof of that?

3. Is starting material-constraints the same as the goal constraints?

4. Can you point to where in the paper you discuss how you learned the goal-conditioned cost network offline (mentioned in the abstract).

**Limitations:**

Authors discuss limitations and those are not concerning (re societal impact, etc).

---

> ### Author Rebuttal · Authors · 2024-08-07
>
> Thank you for your feedback and questions, they bring up some interesting discussion points that will help improve our work!
>
> ---
>
> **Reviewer:** _Proof of soundness and completeness of algorithm 1..._
>
> **Response:**
>
> We agree that some detail regarding algorithm 1 is missing and will add the following justification in the final manuscript.
>
> The purpose of Algorithm 1 is to add the $N$ “best” forward reactions to add to the search graph given a particular reactant $m_1$ from which we want to eventually reach some target molecule $m_2$. If we had access to the complete reaction graph $\mathcal{G}$, we could find the $N$ reactions that use $m_1$ as a reactant that minimize the total cost to reach $m_2$ in $\mathcal{G}$. Thus, $f_t(m_1, m_2)$ would yield the set of templates corresponding to these $N$ reactions, and for each of the bidirectional templates $t_i$, $f_b(m_1, m_2, t_i)$ would yield the missing reactant $m_{3,i}$ in the reaction. We instead approximate $\mathcal{G}$ with the partial reaction graph $G_{\text{FWD}}$ and train neural networks to predict the best templates and building blocks using the reactions in $G_{\text{FWD}}$. Thus, to perform a forward expansion in DESP on $m_1$ conditioned on $m_2$, we approximate $f_t$ with a trained forward template model and return the $N$ most likely templates to apply to $m_1$. For each template, if the template is unimolecular, we can simply apply the template $m_1$ and add the reaction to the search graph. For bimolecular templates, we instead use a building block model that approximates $f_b$ to predict the missing building block $m_3$. The model predicts the fingerprint of $m_3$, and we perform a $k$-nearest neighbors search on our building block set $\mathcal{B}$ to retrieve the $k$ best building blocks to use as $m_3$. Finally, we apply the bimolecular template to $m_2$ and each retrieved building block and add the reaction to the search graph.
>
> The forward expansion algorithm relies on some limiting assumptions, which we touch on in the paper. Particularly, we ignore the existence of $\geq$3-component reactions, and assume that the missing reactant $m_3$ is a member of $\mathcal{B}$, thus precluding “convergent” bottom-up synthesis plans.
>
> ---
>
> **Reviewer:** _...I fail to see its broader impact to the community..._
>
> **Response:**
>
> We would argue that a novel contribution to synthesis planning is in itself a contribution of broad significance to the scientific community. Synthesis planning is a crucial component in the formulation and discovery of most of our medicines, as well as desirable agrochemicals and materials used across a wide swath of industries. Anecdotally, we have received interest in the algorithm from chemists, chemical engineers, and pharmaceutical companies.
>
> ---
>
> **Reviewer:** _Why Bi-directional search?..._
>
> **Response:**
>
> Thank you for the constructive feedback and references! Our implementations of the unidirectional search baselines (including our top-down guided ablation, Retro* + D) do define the starting material goal constraints, and we posited that bidirectional search may be well suited for affording efficiency gains on this task. We find that DESP outperforms all baseline methods on the task, and the ablation study demonstrates that bidirectional search contributes to the improved performance. We believe that these results provide a solid empirical case for the utility of bidirectional search in synthesis planning. Though there are many exciting avenues for improvement of either unidirectional or bidirectional search with preferences and/or constraints, we view such explorations as out of the scope of this paper.
>
> ---
>
> **Reviewer:** _Could there be multiple solutions..._
>
> **Response:**
>
> Thank you for raising this question. For a given target, it is essentially guaranteed that there are many solutions to the general synthesis planning problem. In spite of this, the problem is intractable for many targets without the use of ML or heuristics to prune the search space, so we frame the problem as efficiently finding _some_ solution rather than finding the optimal solution. In any case, knowing that each reaction step is costly in the real world motivates our evaluation in **Table 3** of average number of reaction steps as an indicator of quality / cost. Our heuristic cost networks are also trained with this notion of cost in mind--the training labels correspond to the optimal number of reaction steps (to reach the specified goal molecule) based on the reaction corpus.
>
> A limitation of DESP is that the first route found is not necessarily the cost-optimal route. First, each end of our search is a variant of Retro*. As proven in [1], the unidirectional search only guarantees an optimal solution on termination with admissible heuristics, which our neural networks do not provide. Second, even with an admissible heuristic, bidirectional A* search does not guarantee optimality upon meeting [2].
>
> We emphasize that chemists generally care more about obtaining multiple reasonable routes quickly than obtaining the theoretically optimal route, and our evaluations are designed accordingly. Nevertheless, we will summarize the discussed limitations and outlook in our final manuscript.
>
> ---
>
> **Reviewer:** _Is starting material-constraints the same as the goal constraints?_
>
> **Response:**
> Yes, thank you for raising this question; we will make an effort to be more clear / consistent about such terminology in the final version.
>
> ---
>
> **Reviewer:** _Can you point to where in the paper you discuss how you learned the goal-conditioned cost network offline..._
>
> **Response:**
>
> **Section 3.4** discusses the high level procedure and loss function, while details (including **Figure 5**) can be found in **Section A.4**.
>
> ---
>
> [1] Chen, B. et al. "Retro*: Learning Retrosynthetic Planning with Neural Guided A* Search". _ICML_ (2020).
>
> [2] Kaindl & Kainz. "Bidirectional Heuristic Search Reconsidered". _JAIR_ (1997).

---

> > ### Comment · Reviewer_FuFE · 2024-08-08
> > **Reply to rebuttal**
> >
> > Thank you for answering my questions. I think more details on Algorithm 1 is definitely needed even the full proof of soundness and correctness.

---

> > > ### Author Response · Authors · 2024-08-09
> > > **Proof of soundness and correctness**
> > >
> > > Thank you for your response. We provide a proof of soundness and completeness below.
> > >
> > > ---
> > >
> > > Assume that we have access to all unimolecular and bimolecular reactions that involve at most one non-buyable (forming $G_{\text{FWD}}$), giving us access to target functions $f_t$ and $f_b$ (as described in our response). We prove that **Algorithm 1** is sound and complete for $N = 1$ and $K = 1$.
> > >
> > > Consider the set of routes $S$ for which $m_1$ is a leaf node and $m_2$ is a root and no such route in  $G_{\text{FWD}}$ has lower total cost. On input $m_1, m_2$, the algorithm is sound if the algorithm only adds a reaction $R$ involving $m_1$ if $R$ is in some route in $S$. We call such $R$ an _optimal reaction_. Additionally, the algorithm is complete if it adds an optimal reaction for any input. Since we know $f_t$ and $f_b$, our algorithm computes $f_t(m_1, m_2)$ (and $f_b(m_1, m_2, f_t(m_1, m_2))$ if necessary) which comprises an optimal reaction by the definition of the functions. Thus, the algorithm is sound and complete.
> > >
> > > ---
> > >
> > > In practice, we do not have $f_t$ and $f_b$ and instead approximate them with neural networks, meaning that the algorithm is not sound or complete. This applies to **any** expansion policy of an existing CASP tool and justifies the use of search with higher branching ratios ($N$ and $K$), as the expansion policy will only sometimes give the “optimal reaction” as its top-1 output. Thus, we believe that the provided proof is somewhat tautological and does not significantly enhance our contributions. In any case, we will provide more detail about **Algorithm 1** into our final version, incorporating our initial response and the discussed properties of CASP expansions.

---

### Official Review · Reviewer_pGH2 · 2024-07-11

**Soundness:** 3
**Presentation:** 3
**Contribution:** 3
**Rating:** 7
**Confidence:** 4

**Summary:**

The paper considers computer aided synthesis planning with applications to retrosynthetic analysis in chemistry where the goal is to find a reaction route from purchasable materials to a target molecule. The latter is an important problem with real applications in areas such as drug discovery. In the current landscape of algorithms for retrosynthetic analysis the common practice is to assume reachability to arbitrary materials thus failing to address an important constraint where using specific molecules is most desired. Therefore, the paper proposes a bi-directional graph search algorithm that is restricted to using a user specified set of purchasable basic materials. The algorithm called DESP is guided by a heuristic function derived from a goal-conditioned cost network (neural network) learned offline from a set of valid chemical reactions. The empirical evaluation is carried out on standard benchmarks for retrosynthetic analysis. The results show clearly that the new bi-directional search algorithm improves considerably over existing state-of-the-art algorithms for this domain.

**Strengths:**

- The paper considers an important problem in AI and develops a new more efficient search algorithm to solve it optimally. Specifically, bi-directional search (although a well established search method) appears to be a novel application to retrosynthetic analysis.

- The paper is well written and organised. It is therefore relatively easy to follow. The quality of the presentation is overall quite good.

- The experimental evaluation is sound and covers well most of the prior work in terms of competing algorithms. The results are presented in a relatively clear manner and therefore it is fairly easy to understand the performance gains achieved by the proposed method.

**Weaknesses:**

I was not able to find any major weakness in this paper. Perhaps including a small numerical example to show more clearly how the node values are computed and updated during search would help improve the quality of the presentation.

**Questions:**

- What is dn() and rn() refering to in Equations 3-6? Are these values related to the proof and disproof numbers from the A* algorithm proposed by [Kishimoto et al, 2019]?

**Limitations:**

The limitations of the proposed method are discussed fairly clearly in the paper.

---

> ### Author Rebuttal · Authors · 2024-08-07
>
> Thank you for your review and positive reception of our paper! Addressing your comments:
>
> ---
>
> **Reviewer:**
>
> _I was not able to find any major weakness in this paper. Perhaps including a small numerical example to show more clearly how the node values are computed and updated during search would help improve the quality of the presentation._
>
> **Response:**
>
> This is a great suggestion! We have created such a figure (**Figure 1** in author rebuttal) and will include it in the final version of the paper.
>
> ---
>
> **Reviewer:**
>
> _What is dn() and rn() refering to in Equations 3-6? Are these values related to the proof and disproof numbers from the A* algorithm proposed by [Kishimoto et al, 2019]?_
>
> **Response:**
>
> Thank you for the question. $rn(m|G)$ is actually a quantity directly from the Retro* algorithm of Chen et al. [1]. It represents the “reaction number,” the total estimated cost of synthesizing a particular molecule $m$ based on the current search graph. $dn(m|G)$ is an analogous quantity that we developed in order to incorporate the guidance from the synthetic distance values as well. $rn(m|G)$ is more specifically described in **Section A.2**, and $dn(m|G)$ can be better understood by our explanation in **Section A.5**, as well as the new **Figure 1** we have included in the author rebuttal. We will make sure these quantities are clearly described in the final version of the paper.
>
> ---
>
> [1] Chen, B. et al. "Retro*: Learning Retrosynthetic Planning with Neural Guided A* Search". _ICML_ (2020).

---

> > ### Comment · Reviewer_pGH2 · 2024-08-10
> >
> > Thanks for the clarifications.

---

### Official Review · Reviewer_WuVQ · 2024-07-14

**Soundness:** 4
**Presentation:** 4
**Contribution:** 4
**Rating:** 8
**Confidence:** 3

**Summary:**

This paper proposes a bidirectional search algorithm for chemical synthesis, in the case where we also have certain "part of the way there" molecules that we would like to include in the discovered synthesis route.

**Strengths:**

The paper is very clearly written, with lots of great notation and detail provided. Despite not being familiar with this area, I found the high level ideas easy to follow.

I am familiar with top-down and bottom up program synthesis, so I can follow the high level ideas. But I am not very knowledgable about chemical synthesis and the literature on it. I did not closely read the details of the algorithm, other than trying to understand the high level additions compared with the Retro* baseline. The fact that I could still get a lot of technical value from the paper, despite not paying attention to the chemistry details, and thinking of it just as a generic synthesis-y problem, is to be commended and a sign of the paper's strength and significance.

The results look good.

Bidirectional search seems useful to apply to chemical synthesis, especially to the problem studied.

it really seems like great technical caliber has been applied to this problem. Care and precision is applied to creating the algorithm.

**Weaknesses:**

By adding bidirectional search, the algorithm gets a bit complicated. Due to my lack of familiarity with the area, I'm not 100% sure how to weigh the complexity vs performance tradeoff, but I am inclined to ignore it given how clearly the algorithm is described and how easy the high level motivation for it is.

**Questions:**

- Is the BFS top-down only or bottom-up only? Would it be useful to include a "bidirectional BFS" baseline?

Suggestions/typos:
- I would suggest including a high level explanation of how Retro* compares with DESP in the main text . it sounds like Retro* is top-down only, and DESP = Retro* + synthetic distance + bottom up expansions?

- it might be good to include more details about the compared baselines during the paper. In particular, what is Retro*+$D$? the appendix only describes Retro*. This is almost certainly clear to someone who reads through all the details of the approaches and understands them (Retro* + D just adds the synthetic distance D to Retro*, somehow based on how Retro* and the current work works, and the appendix probably meant ...)  but to clarify, the appendix describes Retro* as DESP without bottom-up expansions. So, at a high level, is Retro* just top-down only, and  DESP is bidirectional, plus the synthetic distance?
- line 106 typo, two front-to-end's

You should check out this reference: https://dl.acm.org/doi/pdf/10.1145/3571226. They use "cuts", which I believe are similar in that using certain "cuts" lets you stake out some target in the "middle" of the search and then split the synthesis problem into two problems, one of reaching the midpoint, and then going from the midpoint to the target. (I haven't read this paper in a while, and sort of forget it, so I might be wrong)
- Maybe this is relevant to addressing the convergent synthesis limitation? not sure.

**Limitations:**

The limitations section looks great!

---

> ### Author Rebuttal · Authors · 2024-08-07
>
> Thank you for your review and the kind words about our work! We address your comments one by one.
>
> ---
>
> **Reviewer:**
>
> _By adding bidirectional search, the algorithm gets a bit complicated. Due to my lack of familiarity with the area, I'm not 100% sure how to weigh the complexity vs performance tradeoff, but I am inclined to ignore it given how clearly the algorithm is described and how easy the high level motivation for it is._
>
> **Response:**
>
> Thank you for pointing this out. We agree that the bidirectional search introduces complexity, though we would argue that some amount of increased complexity is inherent (and not unnecessary) to a double-ended search. One area to tackle the increased complexity is the forward expansion policy (**Algorithm 1**), which is more complex than the retro expansion policy (**Algorithm 2**), as some templates necessitate a second model call and k-NN search over a large database. Synthesizable molecular generation is rapidly attracting more attention ([1], [2], [3] published since we submitted this paper), and we believe a simpler or more elegant method of performing the bottom-up search can likely be integrated in future work. While it cannot trivially be plugged into DESP, ChemProjector (Luo et al. [1]), for instance, demonstrated that a lone transformer architecture can improve bottom-up planning performance by directly decoding the actions to perform, comprising an arguably less complex overall algorithm.
>
> ---
>
> **Reviewer:**
>
> _Is the BFS top-down only or bottom-up only? Would it be useful to include a "bidirectional BFS" baseline?_
>
> **Response:**
>
> The BFS is top-down only. The new baseline is a great idea! We have performed the bidirectional BFS experiment and attached the updated results table in the author rebuttal (**Table 3**). We find that the bidirectional BFS results in fewer node expansions on average than uni-directional BFS, but does not consistently outperform uni-directional baselines like MCTS or Retro*. Thank you for the suggestion.
>
> ---
>
> **Reviewer:**
>
> _I would suggest including a high level explanation of how Retro* compares with DESP in the main text . it sounds like Retro* is top-down only, and DESP = Retro* + synthetic distance + bottom up expansions?_
>
> **Response:**
>
> Thank you for the suggestion. Your understanding is correct, and we will make this more explicit in the final version!
>
> ---
>
> **Reviewer:**
>
> _it might be good to include more details about the compared baselines during the paper. In particular, what is Retro*+D? the appendix only describes Retro*. This is almost certainly clear to someone who reads through all the details of the approaches and understands them (Retro* + D just adds the synthetic distance D to Retro*, somehow based on how Retro* and the current work works, and the appendix probably meant ...) but to clarify, the appendix describes Retro* as DESP without bottom-up expansions. So, at a high level, is Retro* just top-down only, and DESP is bidirectional, plus the synthetic distance?_
>
> **Response:**
>
> Yes, your understanding is correct. “Retro* + D” is DESP as described in **Section 3.2** but without any bottom-up expansions to serve as an ablation. As with our last response, we will make these distinctions more clear in **Section 4.1**. We have also included **Table 2** in the author rebuttal to summarize the evaluated algorithms and their components. We will include this in the appendix of the final manuscript and hope this helps with some of the points of confusion.
>
> ---
>
> **Reviewer:**
>
> _line 106 typo, two front-to-end's_
>
> **Response:**
>
> Thank you for pointing this out! We will fix that.
>
> ---
>
> **Reviewer:**
>
> _You should check out this reference: https://dl.acm.org/doi/pdf/10.1145/3571226. They use "cuts", which I believe are similar in that using certain "cuts" lets you stake out some target in the "middle" of the search and then split the synthesis problem into two problems, one of reaching the midpoint, and then going from the midpoint to the target. (I haven't read this paper in a while, and sort of forget it, so I might be wrong) Maybe this is relevant to addressing the convergent synthesis limitation? not sure._
>
> **Response:**
>
> Thank you for the reference; it brings to mind some very interesting starting points for future work and seems related to ideas we’ve been thinking about as well. The divide-and-conquer or “middle-out” approach sounds like an intriguing method for synthesis planning and would likely naturally extend from DESP. Using something like a “cut” to break a synthesis planning task for a particularly complex molecule into subgoals (i.e. by predicting a key intermediate) also intuitively seems like a promising avenue. We will incorporate these general future directions and the reference into **Section A.8**!
>
> ---
>
> [1] Luo, S., Gao, W., et al. “Projecting molecules into synthesizable chemical spaces” _ICML_ (2024).
>
> [2] Koziarski, M., et al. "RGFN: Synthesizable Molecular Generation Using GFlowNets" _arXiv:2406.08506_ (2024).
>
> [3] Guo, J. & Schwaller, P. "Directly Optimizing for Synthesizability in Generative Molecular Design using Retrosynthesis Models" _arXiv:2407.12186v1_ (2024).

---

> > ### Comment · Reviewer_WuVQ · 2024-08-07
> >
> > Thank you for your response to each of my questions and comments! I have no further questions and will maintain my score of 8.

---

### Official Review · Reviewer_qRkp · 2024-07-31

**Soundness:** 4
**Presentation:** 3
**Contribution:** 3
**Rating:** 7
**Confidence:** 4

**Summary:**

the authors propose a new algorithm for double ended synthesis planning. while this problem has received attention from the community a few decades ago, this has recently not been studied at all.

**Strengths:**

- addresses an important outstanding problem in the community
- sound experimentation from the chemistry perspective

**Weaknesses:**

- none really

**Questions:**

- results from the PAroutes paper and recent work by Tripp et al and Maziarz et al indicate that there are signficantly less or even no differences between the different uni-directional search algorithms than prior work (Chen et al; Retro*) imply. My suggestion would be more clearly flag which baseline implementation of the various algorithms used in the maintext, and also give the used hyperparams for all search algorithms.

---

> ### Author Rebuttal · Authors · 2024-08-07
>
> Thank you for your review! We address your comments and suggestions as follows:
>
> ---
>
> **Reviewer:**
>
> _results from the PAroutes paper and recent work by Tripp et al and Maziarz et al indicate that there are signficantly less or even no differences between the different uni-directional search algorithms than prior work (Chen et al; Retro*) imply._
>
> **Response:**
>
> Thank you for bringing this up. Though these papers perform evaluations on a different problem setting and with different test sets (other than USPTO-190), our results on the starting material-constrained task also find relatively small differences between the two widely used uni-directional search algorithms (MCTS and Retro*) on both USPTO-190 and our new benchmark sets. We will make note of this in the final version of the manuscript.
>
> ---
>
> **Reviewer:**
>
> _My suggestion would be more clearly flag which baseline implementation of the various algorithms used in the maintext, and also give the used hyperparams for all search algorithms._
>
> **Response:**
>
> Thank you for the suggestion! While we do outline the various baselines and relevant hyperparameters in the “Multi-step algorithms” paragraph of **Section 4.1**, we will also add a table to summarize descriptions of our implementations and algorithm-specific hyperparameters to make this more clear. We have included such tables in the author rebuttal (**Tables 1 and 2**) and will incorporate them in our final manuscript.

---

> > ### Comment · Reviewer_qRkp · 2024-08-08
> >
> > Thank you! Again: great paper!

---

### Author Rebuttal · Authors · 2024-08-07

We thank all reviewers for their thoughtful feedback and comments. We have addressed stated weaknesses, questions, and comments in the individual rebuttals. We also include a single page PDF containing additional tables and a figure as referenced in the rebuttals.

---

### Decision · Program_Chairs · 2024-09-25

**Decision:**

Accept (spotlight)

**Comment:**

The authors propose a new algorithm for double-ended chemical synthesis planning,
where the goal is to find a reaction route from purchasable materials to a target molecule.
The paper is well-written and presents significant results for an important problem
with applications in drug synthesis.